# Learning to Predict Without Looking Ahead: World Models Without Forward Prediction

**C. Daniel Freeman, Luke Metz, David Ha**
Google Brain
{cdfreeman, lmetz, hadavid}@google.com

## Abstract

Much of model-based reinforcement learning involves learning a model of an agent's world, and training an agent to leverage this model to perform a task more efficiently. While these models are demonstrably useful for agents, every naturally occurring model of the world of which we are aware—e.g., a brain—arose as the byproduct of competing evolutionary pressures for survival, not minimization of a supervised forward-predictive loss via gradient descent. That useful models can arise out of the messy and slow optimization process of evolution suggests that forward-predictive modeling can arise as a side-effect of optimization under the right circumstances. Crucially, this optimization process need not explicitly be a forward-predictive loss. In this work, we introduce a modification to traditional reinforcement learning which we call *observational dropout*, whereby we limit the agents ability to observe the real environment at each timestep. In doing so, we can coerce an agent into *learning* a world model to fill in the observation gaps during reinforcement learning. We show that the emerged world model, while not explicitly trained to predict the future, can help the agent learn key skills required to perform well in its environment. Videos of our results available at https://learningtopredict.github.io/

## 1 Introduction

Much of the motivation of model-based reinforcement learning (RL) derives from the potential utility of learned models for downstream tasks, like prediction [13, 15], planning [1, 35, 40, 41, 43, 64], and counterfactual reasoning [9, 28]. Whether such models are learned from data, or created from domain knowledge, there's an implicit assumption that an agent's *world model* [21, 52, 66] is a forward model for predicting future states. While a *perfect* forward model will undoubtedly deliver great utility, they are difficult to create, thus much of the research has been focused on either dealing with uncertainties of forward models [11, 16, 21], or improving their prediction accuracy [22, 28]. While progress has been made with current approaches, it is not clear that models trained explicitly to perform forward prediction are the only possible or even desirable solution.

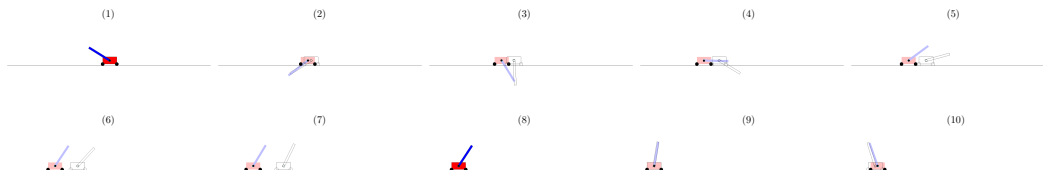

Figure 1: Our agent is given only infrequent observations of its environment (e.g., frames 1, 8), and must learn a world model to fill in the observation gaps. The colorless cart-pole represents the predicted observations seen by the policy. Under such constraints, we show that world models can emerge so that the policy can still perform well on a swing-up cart-pole environment.

We hypothesize that explicit forward prediction is not required to learn useful models of the world, and that prediction may arise as an emergent property if it is useful for an agent to perform its task. To encourage prediction to emerge, we introduce a constraint to our agent: at each timestep, the agent is only allowed to observe its environment with some probability $p$. To cope with this constraint, we give our agent an internal model that takes as input both the previous observation and action, and it generates a new observation as an output. Crucially, the input observation to the model will be the ground truth only with probability $p$, while the input observation will be its previously generated one with probability $1 - p$. The agent's policy will act on this internal observation without knowing whether it is real, or generated by its internal model. In this work, we investigate to what extent world models trained with policy gradients behave like forward predictive models, by restricting the agent's ability to observe its environment.

By jointly learning both the policy and model to perform well on the given task, we can directly optimize the model without ever explicitly optimizing for forward prediction. This allows the model to focus on generating any "predictions" that are useful for the policy to perform well on the task, even if they are not realistic. The models that emerge under our constraints capture the essence of what the agent needs to see from the world. We conduct various experiments to show, under certain conditions, that the models learn to behave like imperfect forward predictors. We demonstrate that these models can be used to generate environments that do not follow the rules that govern the actual environment, but nonetheless can be used to teach the agent important skills needed in the actual environment. We also examine the role of inductive biases in the world model, and show that the architecture of the model plays a role in not only in performance, but also interpretability.

## 2 Related Work

One promising reason to learn models of the world is to accelerate learning of policies by training these models. These works obtain experience from the real environment, and fit a model directly to this data. Some of the earliest work leverage simple model parameterizations – e.g. learnable parameters for system identification [46]. Recently, there has been large interest in using more flexible parameterizations in the form of function approximators. The earliest work we are aware of that uses feed forward neural networks as predictive models for tasks is Werbos [66]. To model time dependence, recurrent neural network were introduced in [52]. Recently, as our modeling abilities increased, there has been renewed interest in directly modeling pixels [22, 29, 45, 59]. Mathieu et al. [37] modify the loss function used to generate more realistic predictions. Denton and Fergus [12] propose a stochastic model which learns to predict the next frame in a sequence, whereas Finn et al. [15] employ a different parameterization involving predicting pixel movement as opposed to directly predicting pixels. Kumar et al. [32] employ flow based tractable density models to learn models, and Ha and Schmidhuber [21] leverages a VAE-RNN architecture to learn an embedding of pixel data across time. Hafner et al. [22] propose to learn a latent space, and learn forward dynamics in this latent space. Other methods utilize probabilistic dynamics models which allow for better planning in the face of uncertainty [11, 16]. Presaging much of this work is [57], which learns a model that can predict environment state over multiple timescales via imagined rollouts.

As both predictive modeling and control improves there has been a large number of successes leveraging learned predictive models in Atari [8, 28] and robotics [14]. Unlike our work, all of these methods leverage transitions to learn an explicit dynamics model. Despite advances in forward predictive modeling, the application of such models is limited to relatively simple domains where models perform well.

Errors in the world model compound, and cause issues when used for control [3, 62]. Amos et al. [2], similar to our work, directly optimizes the dynamics model against loss by differentiating through a planning procedure, and Schmidhuber [51] proposes a similar idea of improving the internal model using an RNN, although the RNN world model is initially trained to perform forward prediction. In this work we structure our learning problem so a model of the world will emerge as a result of solving a given task. This notion of emergent behavior has been explored in a number of different areas and broadly is called "representation learning" [6]. Early work on autoencoders leverage reconstruction based losses to learn meaningful features [26, 33]. Follow up work focuses on learning "disentangled" representations by enforcing more structure in the learning procedure[24, 25]. Self supervised approaches construct other learning problems, e.g. solving a jigsaw puzzle [42], or leveraging temporal structure [44, 56]. Alternative setups, closer to our own specify a specific

learning problem and observe that by solving these problems lead to interesting learned behavior (e.g. grid cells) [4, 10]. In the context of learning models, Watter et al. [65] construct a locally linear latent space where planning can then be performed.

The force driving model improvement in our work consists of black box optimization. In an effort to emulate nature, evolutionary algorithms where proposed [18, 23, 27, 60, 67]. These algorithms are robust and will adapt to constraints such as ours while still solving the given task [7, 34]. Recently, reinforcement learning has emerged as a promising framework to tackle optimization leveraging the sequential nature of the world for increased efficiency [38, 39, 53, 54, 61]. The exact type of the optimization is of less importance to us in this work and thus we choose to use a simple population-based optimization algorithm [68] with connections to evolution strategies [47, 50, 55].

The boundary between what is considered *model-free* and *model-based* reinforcement learning is blurred when one can considers both the model network and controller network together as one giant policy that can be trained end-to-end with model-free methods. [49] demonstrates this by training both world model and policy via evolution. Earlier works [17, 36] demonstrate that agents can learn goal-directed internal models by delaying or omitting sensory information. Instead of performance, however, this work focus on understanding what these models learn and show there usefulness – e.g. training a policy inside the learned models.

## 3   Motivation: When a random world model is good enough

A common goal when learning a world model is to learn a perfect forward predictor. In this section, we provide intuitions for why this is not always necessary, and demonstrate how learning on random "world models" can lead to performant policies when transferred to the real world. For simplicity, we consider the classical control task of balance cart-pole[5]. While there are many ways of constructing world models for cart-pole, an optimal forward predictive model will have to generate trajectories of solutions to the simple linear differential equation describing the pole's dynamics near the unstable equilibrium point[1]. One particular coefficient matrix fully describes these dynamics, thus, for this example, we identify this coefficient matrix as the free parameters of the world model, $M$.

While this unique $M$ perfectly describe the dynamics of the pole, if our objective is only to stabilize the system—*not* achieve perfect forward prediction—it stands to reason that we may not necessarily need to know these exact dynamics. In fact, if one solves for the linear feedback parameters that stabilize a cart-pole system with coefficient matrix $M'$ (not necessarily equal to $M$), for a wide variety of $M'$, those same linear feedback parameters will also stabilize the "true" dynamics $M$. Thus one successful, albeit silly strategy for solving balance cart-pole is choosing a random $M'$, finding linear feedback parameters that stabilize this $M'$, and then deploying those same feedback controls to the "real" model $M$. We provide the details of this procedure in the Appendix.

Note that the *world model* learned in this way is almost arbitrarily wrong. It does not produce useful forward predictions, nor does it accurately estimate any of the parameters of the *real* world like the length of the pole, or the mass of the cart. Nonetheless, it can be used to produce a successful stabilizing policy. In sum, this toy problem exhibits three interesting qualities: **1.** That a world model can be learned that produces a valid policy without needing a forward predictive loss, **2.** That a world model need not itself be forward predictive (at all) to facilitate finding a valid policy, and **3.** That the inductive bias intrinsic to one's world model almost entirely controls the ease of optimization of the final policy. Unfortunately, most real world environments are not this simple and will not lead to performant policies without ever observing the real world. Nonetheless, the underlying lesson that a world model can be quite wrong, so long as it is wrong the in the *right* way, will be a recurring theme.

## 4   Emergent world models by learning to fill in gaps

In the previous section, we outlined a strategy for finding policies without even "seeing" the real world. In this section, we relax this constraint and allow the agent to periodically switch between real observations and simulated observations generated by a world model. We call this method *observational dropout*, inspired by  [58].

Mechanistically, this amounts to a map between a single markov decision process (MDP) into a different MDP with an augmented state space. Instead of only optimizing the agent in the real environment, with some probability, at every frame, the agent uses its internal world model to produce an observation of the world conditioned on its previous observation. When samples from the real world are used, the state of the world model is reset to the real state— effectively resynchronizing the agent's model to the real world.

To show this, consider an MDP with states $s \in \mathcal{S}$, transition distribution $s^{t+1} \sim P(s^t, a^t)$, and reward distribution $R(s^t, a, s^{t+1})$ we can create a new partially observed MDP with 2 states, $s' = (s_{orig}, s_{model}) \in (\mathcal{S}, \mathcal{S})$, consisting of both the original states, and the internal state produced by the world model. The transition function then switches between the real, and world model states with some probability $p$:

$$P'(a^t, (s')^t) = \begin{cases} (s^{t+1}_{orig}, s^{t+1}_{orig}), & \text{if } p < r \\ (s^{t+1}_{orig}, s^{t+1}_{model}), & \text{if } p \geq r \end{cases} \tag{1}$$

where $r \sim \text{Uniform}(0, 1)$, $s^{t+1}_{orig}$ is the real environment transition, $s^{t+1}_{orig} \sim P(s^t_{orig}, a^t)$, $s^{t+1}_{model}$ is the next world model transition, $s^{t+1}_{model} \sim M(s^t_{model}, a^t; \phi)$, $p$ is the peek probability.

The observation space of this new partially observed MDP is always the second entry of the state tuple, $s'$. As before, we care about performing well on the real environment thus the reward function is the same as the original environment: $R'(s^t, a^t, s^{t+1}) = R(s^t_{orig}, a^t, s^{t+1}_{orig})$. Our learning task consists of training an agent, $\pi(s; \theta)$, and the world model, $M(s, a^t; \phi)$ to maximize reward in this augmented MDP. In our work, we parameterize our world model $M$, and our policy $\pi$, as neural networks with parameters $\phi$ and $\theta$ respectively. While it's possible to optimize this objective with any reinforcement learning method [38, 39, 53, 54], we choose to use population based REINFORCE [68] due to its simplicity and effectiveness at achieving high scores on various tasks [19, 20, 50]. By restricting the observations, we make optimization harder and thus expect worse performance on the underlying task. We can use this optimization procedure, however, to drive learning of the world model much in the same way evolution drove our internal world models.

One might worry that a policy with sufficient capacity could extract useful data from a world model, even if that world model's features weren't easily interpretable. In this limit, our procedure starts looking like a strange sort of recurrent network, where the world model "learns" to extract difficult-to-interpret features (like, e.g., the hidden state of an RNN) from the world state, and then the policy is powerful enough to learn to use these features to make decisions about how to act. While this is indeed a possibility, in practice, we usually constrain the capacity of the policies we studied to be small enough that this did not occur. For a counter-example, see the fully connected world model for the grid world tasks in Section 4.2.

## 4.1 What policies can be learned from world models emerged from observation dropout?

As the balance cart-pole task discussed earlier can be trivially solved with a wide range of parameters for a simple linear policy, we conduct experiments where we apply observational dropout on the more difficult swing up cart-pole—a task that cannot be solved with a linear policy, as it requires the agent to learn two distinct subtasks: (1) to add energy to the system when it needs to swing up the pole, and (2) to remove energy to balance the pole once the pole is close to the unstable, upright equilibrium [63]. Our setup is closely based on the environment described in [16, 69], where the ground truth dynamics of the environment is described as $[\ddot{x}, \ddot{\theta}] = F(x, \theta, \dot{x}, \dot{\theta})$. $F$ is a system of non-linear equations, and the agent is rewarded for getting $x$ close to zero and $cos(\theta)$ close to one. For more details, see the Appendix.[2]

The setup of the cart-pole experiment augmented with observational dropout is visualized in Figure 1. We report the performance of our agent trained in environments with various peek probabilities, $p$, in Figure 2 (left). A result higher than $\sim 500$ means that the agent is able to swing up and balance the cart-pole most of the time. Interestingly, the agent is still able to solve the task even when on looking at a tenth of the frames ($p = 10\%$), and even at a lower $p = 5\%$, it solves the task half of the time.

To understand the extent to which the policy, $\pi$ relies on the learned world model, $M$, and to probe the dynamics learned world model, we trained a new policy entirely within learned world model and then

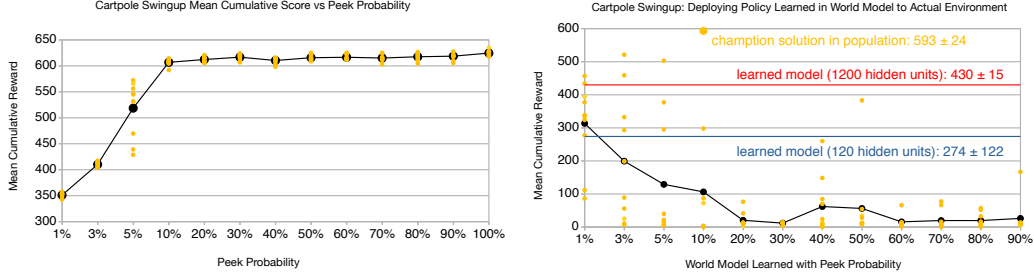

Figure 2: **Left:** Performance of cart-pole swing up under various observational dropout probabilities, $p$. Here, both the policy and world model are learned. **Right:** Performance of deploying policies trained from scratch inside of the environment generated by the world model, in the actual environment. For each $p$, the experiment is run 10 times independently (orange). Performance is measured by averaging cumulative scores over 100 rollouts. Model-based baseline performances learned via a forward-predictive loss are indicated in red, blue. Note how world models learned when trained under approximately 3-5% observational dropout can be used to train performant policies.

deployed these policies back to the original environment. Results in Figure 2 (right). Qualitatively, the agent learns to swing up the pole, and balance it for a short period of time when it achieves a mean reward above $\sim 300$. Below this threshold the agent typically swings the pole around continuously, or navigates off the screen. We observe that at low peek probabilities, a higher percentage of learned world models can be used to train policies that behave correctly under the actual dynamics, despite failing to completely solve the task. At higher peek probabilities, the learned dynamics model is not needed to solve the task thus is never learned.

We have compared our approach to baseline model-based approach where we explicitly train our model to predict the next observation on a dataset collected from training a model-free agent from scratch to solving the task. To our surprise, we find it interesting that our approach can produce models that outperform an explicitly learned model with the same architecture size (120 units) for cart-pole transfer task. This advantage goes away, however, if we scale up the forward predictive model width by 10x.

(a) Policy learned in environment generated using world model.

(b) Deploying policy learned in (a) into real environment.

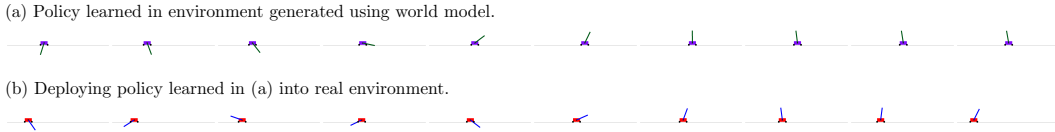

Figure 3: **a.** In the generated environment, the cart-pole stabilizes at an angle that is not perfectly perpendicular, due to its imperfect nature. **b.** This policy is still able to swing up the cart-pole in the actual environment, although it remains balanced only for some time before falling down. The world model is jointly trained with an observational dropout probability of $p = 5\%$.

Figure 3 depicts a trajectory of a policy trained entirely within a learned world model deployed on the actual environment. It is interesting to note that the dynamics in the world model, $M$, are not perfect–for instance, the optimal policy inside the world model can only swing up and balance the pole at an angle that is not perpendicular to the ground. We notice in other world models, the optimal policy learns to swing up the pole and only balance it for a short period of time, even in the self-contained world model. It should not surprise us then, that the most successful policies when deployed back to the actual environment can swing up and only balance the pole for a short while, before the pole falls down.

As noted earlier, the task of stabilizing the pole once it is near its target state (when $x, \theta, \dot{x}, \dot{\theta}$ is near zero) is trivial, hence a policy, $\pi$, jointly trained with world model, $M$, will not require accurate predictions to keep the pole balanced. For this subtask, $\pi$ needs only to occasionally observe the actual world and realign its internal observation with reality. Conversely, the subtask of swinging the pole upwards and then lowering the velocities is much more challenging, hence $\pi$ will rely on the world model to captures the essence of the dynamics for it to accomplish the subtask. The world model $M$ only learns the *difficult* part of the real world, as that is all that is required of it to facilitate the policy performing well on the task.

## 4.2 Examining world models' inductive biases in a grid world

To illustrate the generality of our method to more varied domains, and to further emphasize the role played by inductive bias in our models, we consider an additional problem: a classic search / avoidance task in a grid world. In this problem, an agent navigates a grid environment with randomly placed apples and fires. Apples provide reward, and fires provide negative reward. The agent is allowed to move in the four cardinal directions, or to perform a no-op. For more details, please refer to the Appendix.

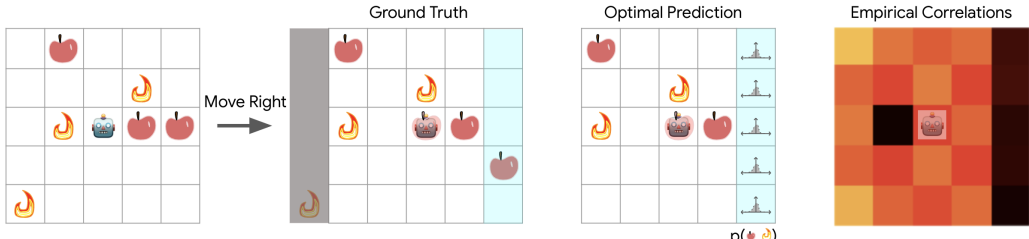

Figure 4: A cartoon demonstrating the shift of the receptive field of the world model as it moves to the right. The greyed out column indicates the column of forgotten data, and the light blue column indicates the "new" information gleaned from moving to the right. An optimal predictor would learn the distribution function $p$ and sample from it to populate this rightmost column, and would match the ground truth everywhere else. The rightmost heatmap illustrates how predictions of a convolutional model correlate with the ground truth (more orange = more predictive) when moving to the right, averaged over 1000 randomized right-moving steps. See the Appendix for more details. Crucially, this heat map is most predictive for the cells the agent can actually see, and is less predictive for the cells right outside its field of view (the rightmost column) as expected.

For simplicity, we considered only stateless policies and world models. While this necessarily limits the expressive capacity of our world models, the optimal forward predictive model within this class of networks is straightforward to consider: movement of the agent essentially corresponds to a bit-shift map on the world model's observation vectors. For example, for an optimal forward predictor, if an agent moves rightwards, every apple and fire within its receptive field should shift to the left. The leftmost column of observations shifts out of sight, and is forgotten—as the model is stateless—and the rightmost column of observations should be populated according to some distribution which depends on the locations of apples and fires visible to the agent, as well as the particular scheme used to populate the world with apples and fires. Figure 4 illustrates the receptive field of the world model.

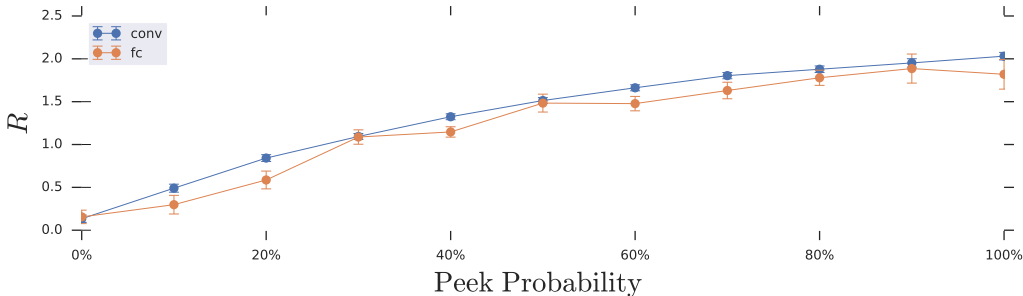

Figure 5: Performance, $R$ of the two architectures, empirically averaged over hundred policies and a thousand rollouts as a function of peek probability, $p$. The convolutional architecture reliably out performs the fully connected architecture. Error bars indicate standard error. Intuitively, a score near 0 amounts to random motion on the lattice—encountering apples as often as fires, and 2 approximately corresponds to encountering apples two to three times more often than fires. A baseline that is trained on a version of the environment without any fires—i.e., a proxy baseline for an agent that can perfectly avoid fires—reliably achieves a score of 3. Agents were trained for 4000 generations.

This partial observability of the world immediately handicaps the ability of the world model to perform long imagined trajectories in comparison with the previous continuous, fully observed cart-pole tasks. Nonetheless, there remains sufficient information in the world to train world models via observational dropout that are predictive.

For our numerical experiments we compared two different world model architectures: a fully connected model and a convolutional model. See the Appendix for more details. Naively, these models are listed in increasing order of inductive bias, but decreasing order of overall capacity (10650 parameters for the fully connected model, 1201 learnable parameters for the convolutional model)—i.e., the fully connected architecture has the highest capacity and the least bias, whereas the convolutional model has the most bias but the least capacity. The performance of these models on the task as a function of peek probability is provided in Figure 5. As in the cart-pole tasks, we trained the agent's policy and world model jointly, where with some probability $p$ the agent sees the ground truth observation instead of predictions from its world model.

Curiously, even though the fully connected architecture has the highest overall capacity, and is capable of learning a transition map closer to the "optimal" forward predictive function for this task if taught to do so via supervised learning of a forward-predictive loss, it reliably performs worse than the convolutional architectures on the search and avoidance task. This is not entirely surprising: the convolutional architectures induce a considerably better prior over the space of world models than the fully connected architecture via their translational invariance. It is comparatively much easier for the convolutional architectures to randomly discover the right sort of transition maps.

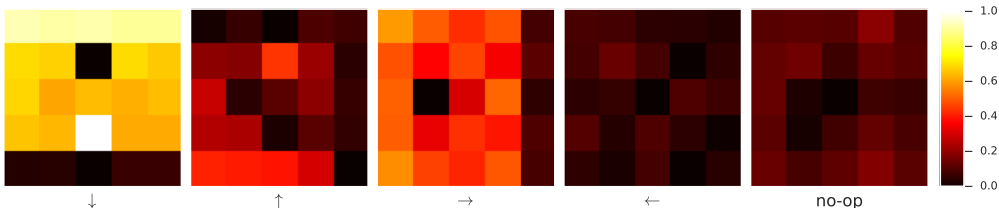

Figure 6: Empirically averaged correlation matrices between a world model's output and the ground truth. Averages were calculated using 1000 random transitions for each direction of a typical convolutional $p = 75\%$ world model. Higher correlation (yellow-white) translates to a world model that is closer to a next frame predictor. Note that a predictive map is not learned for every direction. The row and column, respectively of dark pixels for $\downarrow$ and $\rightarrow$ correspond exactly to the newly-seen pixels for those directions which are indicated in light-blue in Figure 4.

Because the world model is not being explicitly optimized to achieve forward prediction, it doesn't often learn a predictive function for every direction. We selected a typical convolutional world model and plot its empirically averaged correlation with the ground truth next-frames in Figure 6. Here, the world model clearly only learns reliable transition maps for moving down and to the right, which is sufficient. Qualitatively, we found that the convolutional world models learned with peek-probability close to $p = 50\%$ were "best" in that they were more likely to result in accurate transition maps—similar to the cart-pole results indicated in Figure 2 (right). Fully connected world models reliably learned completely uninterpretable transition maps (e.g., see the additional correlation plots in the Appendix). That policies could *almost* achieve the same performance with fully connected world models as with convolutional world model is reminiscent of a recurrent architecture that uses the (generally not-easily-interpretable) hidden state as a feature.

### 4.3 Car Racing: Keep your eyes *off* the road

In more challenging environments, observations are often expressed as high dimensional pixel images rather than state vectors. In this experiment, we apply observation dropout to learn a world model of a car racing game from pixel observations. We would like to know to what extent the world model can facilitate the policy at driving if the agent is only allowed to see the road only only a fraction of the time. We are also interested in the representations the model learns to facilitate driving, and in measuring the usefulness of its internal representation for this task.

In Car Racing [31], the agent's goal is to drive around the tracks, which are randomly generated for each trial, and drive over as many tiles as possibles in the shortest time. At each timestep, the environment provides the agent with a high dimensional pixel image observation, and the agent outputs 3 continuous action parameters that control the car's steering, acceleration, and brakes.

To reduce the dimensionality of the pixel observations, we follow the procedure in [21] and train a Variational Autoencoder (VAE) [30, 48] using on rollouts collected from a random policy, to compress a pixel observation into a small dimensional latent vector $z$. Our agent will use $z$ instead as its observation. Examples of pixel observations, and reconstructions from their compressed

Actual frames from rollout (a)                                                                                time ⟶

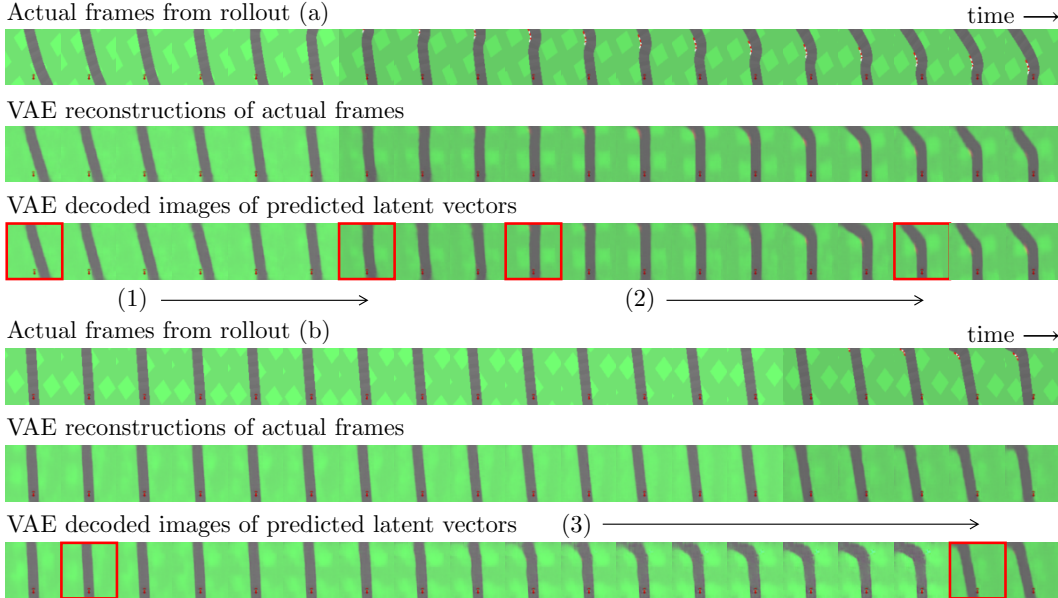

VAE reconstructions of actual frames

VAE decoded images of predicted latent vectors

(1) ⟶                          (2) ⟶

Actual frames from rollout (b)                                                                                time ⟶

VAE reconstructions of actual frames

VAE decoded images of predicted latent vectors          (3) ⟶

Figure 7: Two examples of action-conditioned predictions from a world model trained at $p = 10\%$ (bottom rows). Red boxes indicate actual observations from the environment the agent is allowed to see. While the agent is devoid of sight, the world model predicts (1) small movements of the car relative to the track and (2) upcoming turns. Without access to actual observations for many timesteps, it incorrectly predicts a turn in (3) until an actual observation realigns the world model with reality.

representations are shown in the first 2 rows of Figure 7. Our policy, a feed forward network, will act on actual observations with probability $p$, otherwise on observations produced by the world model.

Our world model, $M$, a small feed forward network with a hidden layer, outputs the change of the mean latent vector $z$, conditioned on the previous observation (actual or predicted) and action taken (i.e $\Delta z = M(z, a)$). We can use the VAE's decoder to visualize the latent vectors produced by $M$, and compare them with the actual observations that the agent is not able to see (Figure 7). We observe that our world model, while not explicitly trained to predict future frames, are still able to make meaningful action-conditioned predictions. The model also learns to predict local changes in the car's position relative to the road given the action taken, and also attempts to predict upcoming curves.

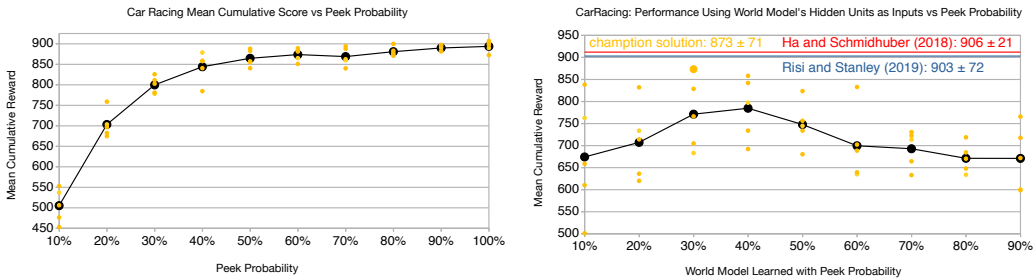

Figure 8: **Left:** Mean performance of Car Racing under various $p$ over 100 trials. **Right:** Mean performance achieved by training a linear policy using only the outputs of the hidden layer of a world model learned at peek probability $p$. We run 5 independent seeds for each $p$ (orange). Model-based baseline performances learned via a forward-predictive loss are indicated in red, blue. We note that in this constrained linear policy setup, our best solution out of a population of trials achieves a performance slightly below reported state-of-the-art results (i.e. [21, 49]). As in the swingup cartpole experiments, the best world models for training policies occur at a characteristic peek probability that roughly coincides with the peek probability at which performance begins to degrade for jointly trained models (i.e., the bend in the left pane occurs near the peak of the right pane).

Our policy $\pi$ is jointly trained with world model $M$ in the car racing environment augmented with a peek probability $p$. The agent's performance is reported in Figure 8 (left). Qualitatively, a score above $\sim 800$ means that the agent can navigate around the track, making the occasional driving error. We see that the agent is still able to perform the task when 70% of the actual observation frames are dropped out, and the world model is relied upon to fill in the observation gaps for the policy.

If the world model produces useful predictions for the policy, then its hidden representation used to produce the predictions should also be useful features to facilitate the task at hand. We can test whether the hidden units of the world model are directly useful for the task, by first freezing the weights of the world model, and then training from scratch a *linear* policy using only the outputs of the intermediate hidden layer of the world model as the only inputs. This feature vector extracted the hidden layer will be mapped directly to the 3 outputs controlling the car, and we can measure the performance of a linear policy using features of world models trained at various peek probabilities.

The results reported in Figure 8 (right) show that world models trained at lower peek probabilities have a higher chance of learning features that are useful enough for a linear controller to achieve an average score of 800. The average performance of the linear controller peaks when using models trained with $p$ around 40%. This suggests that a world model will learn more useful representation when the policy needs to rely more on its predictions as the agent's ability to observe the environment decreases. However, a peek probability too close to zero will hinder the agent's ability to perform its task, especially in non-deterministic environments such as this one, and thus also affect the usefulness of its world model for the real world, as the agent is almost completely disconnected from reality.

## 5   Discussion

In this work, we explore world models that emerge when training with *observational dropout* for several reinforcement learning tasks. In particular, we've demonstrated how effective world models can emerge from the optimization of total reward. Even on these simple environments, the emerged world models do not perfectly model the world, but they facilitate policy learning well enough to solve the studied tasks.

The deficiencies of the world models learned in this way have a consistency: the cart-pole world models learned to swing up the pole, but did not have a perfect notion of equilibrium—the grid world world models could perform reliable bit-shift maps, but only in certain directions—the car racing world model tended to ignore the forward motion of the car, unless a turn was visible to the agent (or imagined). Crucially, none of these deficiencies were catastrophic enough to cripple the agent's performance. In fact, these deficiencies were, in some cases, irrelevant to the performance of the policy. We speculate that the complexity of world models could be greatly reduced if they could fully leverage this idea: that a complete model of the world is actually unnecessary for most tasks—that by identifying the *important* part of the world, policies could be trained significantly more quickly, or more sample efficiently.

We hope this work stimulates further exploration of both model based and model free reinforcement learning, particularly in areas where learning a perfect world model is intractable.

## Acknowledgments

We would like to thank our three reviewers for their helpful comments. Additionally, we would like to thank Alex Alemi, Tom Brown, Douglas Eck, Jaehoon Lee, Błażej Osiński, Ben Poole, Jascha Sohl-Dickstein, Mark Woodward, Andrea Benucci, Julian Togelius, Sebastian Risi, Hugo Ponte, and Brian Cheung for helpful comments, discussions, and advice on early versions of this work. Experiments in this work were conducted with the support of Google Cloud Platform.

## Footnotes

[1]In general, the full dynamics describing cart-pole is non-linear. However, in the limit of a heavy cart and small perturbations about the vertical at low speeds, it reduces to a linear system. See the Appendix for details.

[2]Released code to facilitate reproduction of experiments at https://learningtopredict.github.io/

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
