[Supplementary Material]

## A  Random world models for balance cart-pole

Consider the classical control task of balance cart-pole, where the cart is initialized "close" to the unstable, fully upright equilibrium, and the cart's (not the pole's) acceleration is the only directly controllable parameter.

Following [2], the Lagrangian for cart-pole system takes the form:

$$\mathcal{L} = \frac{1}{2}(M + m)\dot{x}^2 + \frac{1}{2}mL^2\dot{\theta}^2 - mLcos(\theta)\dot{\theta}\dot{x} - mgLcos(\theta) \tag{1}$$

In the presence of a control force, $u(t)$, the solution to the Euler-Lagrange equation $\frac{\partial L}{\partial q} - \frac{\partial}{\partial t}\frac{\partial L}{\partial \dot{q}} = 0$ for $q \in \{\theta, x\}$ yields the full equations of motion:

$$(M + m)\ddot{x} + mLsin(\theta)\dot{\theta}^2 - mLcos(\theta)\ddot{\theta} = u(t) \tag{2}$$

$$mL^2\ddot{\theta} - mLcos(\theta)\ddot{x} - mgLsin(\theta) = 0 \tag{3}$$

Taking the limits that $m \ll M$, $\dot{\theta} \ll 1$, and $\theta \ll 1$—that the pole is light compared to the cart, that the pole is not moving very fast, and that the pole is near the vertical, respectively—the $x$ and $\theta$ components of the differential equation decouple, and the pole dynamics can be rearranged into the matrix equation:

$$\begin{pmatrix} \dot{\theta} \\ \ddot{\theta} \end{pmatrix} = \begin{pmatrix} 0 & 1 \\ \frac{g}{L} & 0 \end{pmatrix} \begin{pmatrix} \theta \\ \dot{\theta} \end{pmatrix} + \begin{pmatrix} 0 \\ \frac{1}{ML} \end{pmatrix} u(t) \tag{4}$$

Finally, the linear feedback ansatz for $u(t)$ is imply:

$$u(t) = \begin{pmatrix} u_1 & u_2 \end{pmatrix} \begin{pmatrix} \theta \\ \dot{\theta} \end{pmatrix} \tag{5}$$

Combining Eq. 4 and Eq. 5 results in 6 as desired.

Linearizing around this equilibrium, taking the linear feedback ansatz for the form of the controller, and considering only the angular degrees of freedom the equations of motion for this system are,

$$\begin{pmatrix} \dot{\theta} \\ \ddot{\theta} \end{pmatrix} = \begin{pmatrix} 0 & 1 \\ \frac{g}{L} + \frac{u_1}{ML} & \frac{u_2}{ML} \end{pmatrix} \begin{pmatrix} \theta \\ \dot{\theta} \end{pmatrix} \sim \begin{pmatrix} a & b \\ c + u_1 & d + u_2 \end{pmatrix} \begin{pmatrix} \theta \\ \dot{\theta} \end{pmatrix} \tag{6}$$

where $\theta$ is the angle of the pole with the vertical, $g$ is the gravitational constant, $L$ is the length of the pole, $M$ is the mass of the cart (considered much larger than the mass of the pole), and $u_1, u_2$ are the free parameters of the controller. Reinterpreted as a model-based reinforcement learning problem, $u_1, u_2$ are the free parameters of a policy $\pi$, and the exact model of the dynamics of the "world" are the solutions to this differential equation. Finding a policy which "solves" Eq. 6—i.e., that drives the pole towards the $\theta = 0$ equilibrium state—is possible via random search. Almost any pair of negative entries stabilizes the pole[1].

We don't often have access to the exact equations of motion for a problem of interest, thus one possible analogous "world-model" version of this task is to find both a policy, $u_1, u_2$, and matrix entries $a, b, c, d$: that when solved, also solve the original task (i.e., Eq. 6 RHS). This task is equivalent to learning a world model for a problem, training a policy entirely within the learned model, and then measuring the transfer of the policy to the real world. Surprisingly, this task is also efficiently

solvable via random search with high probability. Specifically, starting with Gaussian distributed $a, b, c, d$ and then solving for a $u_1^*, u_2^*$ that stabilizes Eq. 6, with probability $p > 0$ those same $u_1^*, u_2^*$ will also stabilize a balance cart-pole problem with $L, g, M \sim O(1)$.

While this cartoon does rely on the simplicity of the solution space for balance cart-pole, it hints at a more general property of learned models for RL tasks: models can be wrong so long as they are wrong in the right way. "Solving" the balance cart-pole task fundamentally amounts to finding $u_1, u_2$ that cause the coefficient matrix to have negative eigenvalues. The class of matrices that is negative definite both when added to $\begin{pmatrix} 0 & 0 \\ u_1 & u_2 \end{pmatrix}$ *and* when $\begin{pmatrix} 0 & 1 \\ \frac{g}{L} + \frac{u_1}{ML} & \frac{u_2}{ML} \end{pmatrix}$ is itself negative definite is large. Thus, looking for $u_1, u_2$ that stabilize random matrices in the neighborhood of the coefficient matrix is a sensible, albeit highly inductively biased, strategy. Of course, most problems do not afford such a dramatic freedom in the dimensionality of the solution manifold.

# B  Experimental Details

Please visit the web version at https://learningtopredict.github.io/ of this paper for information about the released code for reproducing experiments.

Experiments were performed used multi-core machines on Google Cloud Platform, for various peek probability settings, and also for multiple independent runs with different initial random seeds. Cart-pole swing up experiments were performed on multiple 96 core machine, while car racing and grid world experiments were performed on 64 core machines.

Below we describe architecture setup and experimental details for each experiment.

## B.1  Swing up cart-pole

In our experiments, we fine-tuned individual weight parameters for the champion networks found to measure the performance impact of further training. For this, we used population-based REINFORCE, as in Section 6 of [5]. Our specific approach is based on open source `estool` [3] implementation of population-based REINFORCE with default parameter settings, where we use a population size of 384, and had each agent perform the task 16 times with different initial random seeds for swing up cart-pole. The agent's reward signal used by the policy gradient method is the average reward of the 16 rollouts.

In this task, our policy network is a feed forward network with 5 inputs, 1 hidden layer of 10 tanh units, and 1 output for the action. The world model is another feed forward network with 5 inputs, 30 hidden tanh units, and 5 outputs. We experimented with a larger hidden size, and extra hidden layers, but did not see meaningful differences in performance. All models were trained for 10,000 generations.

## B.2  Grid Worlds

For our grid world experiments, we used the same open source population-based optimizer implementation with default parameters, a population size of 8, and a cumulative reward signal averaged over 4 rollouts.

For the fully connected network experiments, the input to the world model was a flattened list of the $5 \times 5 \times 2$ observation binary variables concatenated with the length 5 one-hot action vector. This was passed into a one layer network with 100 hidden units in the hidden layer, and 50 units in the output layer. Predictions were calculated via thresholding: if an output was greater than .5, it was rounded to 1, otherwise it was rounded to 0. All apple and fire locations were predicted simultaneously.

For the convolutional network experiments, we used a convolutional architecture with shared weights, a $3 \times 3$ kernel where the corner entries were forced to be zero (i.e., only the center pixel, and the pixels in the 4 cardinal directions around it were inputs for each receptive field—that is, 5 of the 9 pixels in the $3 \times 3$ receptive field were active), and 100 channels. For each $3 \times 3$ receptive field, the one hot action vector was concatenated to the flattened field, and then processed by the network. The output of each $3 \times 3$ receptive field was 1-dimensional, and we used the same thresholding scheme as for the fully connected networks—i.e., more than .5 was rounded to 1, and less than .5 was rounded to 0. Apple and Fire observations were predicted with the same network.

For both world model architecture experiments, the same policy architecture was used: a simple two-layer fully connected network with a tanh activation after the first layer, 100 units in the first hidden layer, and 32 units in the second hidden layer, and 5 units in the output layer.

All models were trained for 4000 generations, and all models took between 10 and 100 random steps in a $12 \times 12$ environment with 30 apples and 30 fires.

### B.3 Car Racing

As in the swing up cart-pole experiment, we used the same open source population-based optimizer implementation with default parameters, but due to the extra computation time required, we instead use a population size of 64 and the average cumulative reward of 4 rollouts to calculate as the reward signal.

The code and setup of the VAE for the Car Racing task is taken from [4]. We used the pre-trained VAE made available in [1] with a latent size of 16 which trained following the same procedure as [4]. For simplicity, we do not use the RNN world model described in [4] for achieving state-of-the-art results, but instead, we found that removing the VAE noise from the latent vector for the observation improves results, hence in our experiment, from the point of view of the policy, the observed latent vector $z$ is set to the predicted mean of the encoder, $\mu$, from the pre-trained VAE.

In this task, our policy network is a feed forward network with 16 inputs (the latent vector), 1 hidden layer of 10 tanh units, and 3 outputs for the action space. The world model is another feed forward network with 16 inputs, 10 hidden tanh units, and 16 outputs. The 10 hidden units of this world model was used as the input for a simple linear policy in the experiments. All models were trained for 1,000 generations.

## C   The Grid World Environment

The environments are all square grids with impassable walls on the boundary. Apples and Fires are placed randomly, but so that no tile has more than at most 1 Apple or Fire.

When the agent encounters an apple, it does not consume it until it takes an additional step—i.e., the agent sees the apple on the turn that agent encounters it. Consumed apples are removed from the environment. The agent receives 1 point of reward for every step in the environment, 6 reward for every apple it encounters, and -8 reward for every fire it encounters.

Apples and Fires are represented as binary variables in a $d \times d \times 2$ matrix, for grid width $d$. The agent can perform one of 5 actions—movement in the 4 cardinal directions, or a no-op.

## D   Correlations of predictions

Figure 1: Correlation matrices for several sampled convolutional architectures. The dark pixel immediately adjacent to the agent in many of the correlation plots is a result of the agent failing to predict its own consumption of an apple, because the model used was translationally invariant.

Figure 2: Correlation matrices for several sampled fully connected architectures. Note the lack of interpretability of the learned models, even though the policies learned jointly with these world models were fairly performant.

## Footnotes

[1]"Stability", here, is meant in the formal control theoretic sense—i.e. that the coefficient matrix has only negative eigenvalues