[Reviews · NeurIPS 2019]

Reviewer 1



Main Ideas The high-level motivation of this work is to consider alternatives to learning good forward models, which may not be a desirable solution in all cases. The hypothesis is that a predictive model may arise as an emergent property if such prediction were useful for the agent. The authors test this hypothesis by constraining the agent to only observe states at certain timesteps, requiring a model to learn to fill in the gaps. The model was not trained with a forward prediction objective. Connection to Prior Work. The method introduced in this work seem novel in the context of other literature that train forward models. Other work have also attempted to overcome the difficulties with training forward models, such as by using inverse models [1]. The predictron [2] also learns an implicit model as the submission does. Would the authors be able to include a discussion comparing their work with the two above types of approaches ((1) inverse models and (2) implicit models) in the related work section? Quality - strengths: the authors are careful and honest about evluating the strengths and weaknesses of their work. They evaluated their idea on simple easy-to-analyze tasks and also demonstrated the generality of their method on various domains. Clarity - strengths: the paper is very well written and motivated Originality - strengths: the framing of the problem and the method the authors use seems novel Significance - strengths: this work is a proof-of-concept that training an implicit model with observational dropout in some cases is sufficient for learning policies - weakness: one of the appeals of learning a world model is that such models help facilitate generalization to different tasks in the same domain. For example, one task could be to train on car-racing going forwards and test on car-racing going backwards, where a forward model that is trained to predict the next state given the current state and action could presumably handle. However, the implicit model studied in this submission is inherently tied to the training task, and it is unclear whether such implicit models would help with such a generalization. Would the authors be able to provide a thorough experiment analyzing the limits and capabilities of how their implicit model facilitates generalization to unseen tasks? Overall, I like the perspective of this paper and I think it is well written, well-motivated, and thorough. The key drawback I see is the lack of analysis on how well the proposed method fares on generalizing to unseen tasks. This analysis in my opinion is crucial because a large motivation for learning models in the first place is to facilitate such generalization. [1] Pathak, Deepak, et al. "Zero-shot visual imitation." Proceedings of the IEEE Conference on Computer Vision and Pattern Recognition Workshops. 2018. [2] Silver, David, et al. "The predictron: End-to-end learning and planning." Proceedings of the 34th International Conference on Machine Learning-Volume 70. JMLR. org, 2017. UPDATE AFTER REBUTTAL I appreciated that the authors have conducted the experiment comparing their proposed method with a model-based baseline. I think a more thorough analysis of generalization would make the paper much stronger and I believe the significance of this work is more of a preliminary flavor. The paper is executed carefully and the results were consistent with the author's claims, but this is only the first step. I keep my original score, but would agree with Reviewer 3's comment about motivation and urge the authors to provide a more compelling argument through empirical evaluation for why one would want to consider the proposed idea in the camera-ready.

Reviewer 2



The authors set out to explore the idea how much data needs to be observed from the world and how much can be imagined or simulated through ones own forward model in model-based reinforcement learning tasks. Therefore, observational-dropout is introduced, a method that at each time step probabilistically substitutes real world observations with predictions of a learned world model which is optimized to support the learning of key skills on search and avoidance or navigation task during dropout. Through that mechanism a forward prediction model implicitly arises. Results suggest that observational-dropout improves the generalization of a trained world-model. Interestingly, the dropout rate seems to be crucial for successful training. Keeping too much ground truth observations seems to prevent the world model from learning anything as then the world model is not needed for performing the task. Keeping too little ground truth observations also diminishes performance as then the agent is too strongly disconnected from the world and solely relies on its own predictions to perform the task which over time can strongly deviate from the ground truth. There seems to be a sweet spot for the right amount of dropout at which both the performance of the implicit forward predictor and the task performance peak. These results were shown in a race car environment and a grid world like environment in which an agent is tasked to search for or avoid certain items, which will be made publicly available. Originality: The idea of dropping observations to implicitly train a world model in a model-based reinforcement learning setup is novel and original. Quality: Well written, instructive, paper with actionable implications for future research. Clarity: Overall the paper is clear and well written. Significance: The results are significant but have been only tested on toy problems. It would be great to see how the presented findings hold up in more complex scenarios. All in all, I would like to argue for accepting this paper. The paper is very instructive, clearly written and the idea of training a world model implicitly through dropout is novel and original. The results are interesting and applicable to model-based reinforcement learning setups. The major insight and contribution lies in the analysis of how much of the observations are exactly needed or actually should be dropped to train a generalizable forward predicting world model. The submission included code and the test environment will be published. The only caveat of this paper is that the experiments are performed on very simple navigation tasks and more complex control experiments closer to real-world scenarios would have made the results more convincing. Conceptually this is interesting work, however it is hard to tell whether this would apply to scenarios outside of simulation. EDIT - The authors addressed concerns regarding comparison to prior work and generalizability in the rebuttal, and discuss how in their experiments generalization is task-dependent. Assuming authors execute the changes proposed in the rebuttal and put them in the final draft, I would still argue to accept the paper, but would not be upset if it gets rejected.

Reviewer 3



Edit after author response: I appreciate the model-based baseline and the frank discussion. I think the main pending question for this work is whether there is a reason to learn a model this way instead of with an explicit prediction objective. However, I think this is an interesting direction for exploration. Overall this paper has an interesting idea and does some preliminary work to evaluate it. However, at this point the authors show only that it might be possible to do this, not that there is any reason one might want to. This is a clever paper and I would like to read a more complete version. Originality This idea is fairly original, proposing to implicitly learn what is recognizably a predictive model of the environment without a prediction objective. I would be interested to see further exploration of this idea, and in particular a demonstration of cases where it has an advantage over explicit model learning. The most similar work to this is probably "The Predictron: End-To-End Learning and Planning" (Silver et al.) and the authors should include a discussion of the similarities. Quality The main weakness of this work is in its experiments. The results shown in Figs. 2 & 5 seem unimpressive, and this work contains zero comparisons to any other methods or variants of the model. It is unacceptable that it does not include a comparison to explicit model learning. The results in Figs. 4 & 6 are qualitatively interesting but somewhat hard to interpret; they seem to indicate that the model which is learned is only vaguely related to ground-truth prediction. The comparison between architectures in Fig. 5 shows no significant difference; furthermore, without sharing the amount of data used to train the two architectures it is impossible to evaluate (as inductive biases will be washed out with sufficient data). The number of environment steps used may be computable from the number of training generations, etc in the Appendix but should be explicitly stated in the main body. It is also clear that with an expressive policy which is able to distinguish between real and generated observations, there is no reason at all that the implicit "model" should need to make forward predictions at all. In that case the policy as a whole reduces to a recurrent neural network policy. It would be important to include a discussion of this limitation. Clarity The writing quality in this work is high and I enjoyed reading it. However, there are a few details which could use elaboration: - All experiments should include the number of environment samples used. - The observation space for the cartpole example is not explicitly stated. Significance Currently the significance of this paper is low-medium. It has a clever idea but it does not establish it well enough to motivate follow-on work by others.

[Author Response · NeurIPS 2019]

We thank the reviewers for their feedback and comments on the manuscript! Indeed, this work was largely an exploration of the idea: "Can a world model be learned without using a forward-predictive loss?" Owing to the intense research effort spent on learning forward predictive models in the presence of many a flavor of forward-predictive auxiliary losses, we focused our efforts on simple tasks that best illustrated the trade-offs of our exploration. We are assured to see that we have achieved this as all 3 reviewers thought our method was interesting, and that our presentation was clear. While adding more complex test environments would certainly strengthen our work, we assert that the experiments presented here provide a compelling sketch for a different way to train world models.

Because of the relative maturity of the model-based literature, we were initially hesitant to make direct comparisons with model-based baselines because doing so immediately invites a comparison that we did not intend to draw. However, we fully agree that including such a baseline is useful, informative, and, if nothing else, establishes a long term goal for this program. We have now included model-based baselines for the swing-up cartpole and car-racing tasks (see below), and we have launched more detailed baseline experiments for car-racing (suggested by R3) and gridworld.

| Cartpole Swingup | Reward |
| --- | --- |
| Explicitly learned model from data (120 hidden units) | $274 \pm 122$ |
| Explicitly learned model from data (1200 hidden units) | $430 \pm 15$ |
| Champion solution in our population (120 hidden units) | $593 \pm 24$ |

| CarRacing-v0 | Reward |
| --- | --- |
| Ha and Schmidhuber (NeurIPS 2018) | $906 \pm 21$ |
| Risi and Stanley (GECCO 2019) | $903 \pm 72$ |
| Champion solution in our population | $873 \pm 71$ |

The full baselines and discussion will be readily available in the revision. To our surprise, we find it interesting that our approach can produce models that outperform an explicitly learned model with the same architecture size (120 units) for cartpole transfer task, but this advantage goes away if we scale up the learned model size by 10x. For car-racing, only the best solution out of a population of trials gets near the performance of population-based baselines from the literature, but honestly we did not expect our new implicit approach of learning world models to be this close in performance.

With respect to R1, R2 and R3's comments on prior work (especially the Predictron), we of course will add these overlooked references, and provide a discussion of similarities and differences of these works. In particular for the Predictron, both methods are after the same goals: a latent model emerging from optimization. The predictron does this via learning representations, dynamics, and value estimates end-to-end over imagined rollouts of varying lengths. The primary innovation of observational dropout is to recouple the "imagined" rollouts back to the real observation space—i.e., any sequence of transitions within our learned world models always end back on a "real" frame from the environment, with "real" environment steps in the intervening frames. Additionally, where the predictron derives consistency by demanding that its value estimates obey a Bellman equation, we only demand that the learned dynamics model facilitate the learning of a policy. While similar in spirit, we hope the difference here is clear—it's interesting, for example, to consider how a predictron could be combined with observational dropout, where at each step, instead of necessarily seeing another ground-truth frame from the environment, the predictron sees its own 1-step prediction.

Regarding R1 and R2's concerns on generalization (whether to "harder" or "unseen" tasks): we agree that understanding generalization is crucial, and regret that we did not have more space to discuss this point in the manuscript. In sum: for high observational dropout, too much information is lost, and the world model does not have the capability to accurately reconstruct the system due to noise. At low dropout, the world model might as well not be present, because the policy can ignore the occasionally noisy frame. The extent to which the world model can then transfer to "unseen tasks" is dependent on how well the world model fully captures the dynamics of the system. In some sense, world models learned in this way are "bad" at generalizing, because they are trained to only learn the "relevant" dynamics of the task being studied (as in the grid world task, where it only learned transitions in some but not all directions). In another sense, they're "good" at generalizing, because the "relevant" dynamics of the current task might also be the relevant dynamics of "unseen" tasks (i.e., as in balance cartpole, initialized at an angle not seen during training). It is difficult to make a precise statement here, because the quality of the learned world model truly depends on the dynamics being studied.

R3 had an interesting comment regarding the relative power of the policy versus the world model—indeed, a sufficiently powerful policy could extract meaningful information from the world model, even if the world model was not actually forward predictive at all. In this limit, the joint policy + world model system starts looking like a strange sort of recurrent network. We agree with this assessment, and will add a short discussion highlighting this limit. An alternative interpretation is that our procedure is a way of encouraging a recurrent model's internal state to be dynamically relevant/interpretable—first by mediating a policy's observation space via that lossy dynamics module, and then by constraining the policy itself to be simple, so that it is not, in R3's words, "sufficiently powerful". We are absolutely interested in pursuing this interpretation through an information bottleneck lens, but felt it was out of the scope of this work. Finally, we absolutely will add more details on the number of training steps used for the gridworld tasks (an equal number was used for both architectures, to answer R3), as well as details on the observation space of cartpole. We hope this response has addressed the reviewer's concerns, and would appreciate a reevaluation of your scores!

[Meta-Review · NeurIPS 2019]

Interesting work that explores whether world model be learned without using a forward-predictive loss, and providing a novel perspective on model-based reinforcement learning. Introducing a method of 'observational dropout', the paper presents the first step towards demonstrating the feasibility of learning only the salient features needed for a task. The paper rebuttal has baseline comparisons to model based RL, which will be a valuable addition to the paper.